# Tocopherols in Cultivated Apple *Malus* sp. Seeds: Composition, Variability and Specificity

**DOI:** 10.3390/plants12051169

**Published:** 2023-03-03

**Authors:** Paweł Górnaś, Gunārs Lācis, Inga Mišina, Laila Ikase

**Affiliations:** Institute of Horticulture, Graudu 1, LV-3701 Dobele, Latvia

**Keywords:** apple seeds, tocopherol homologues, geographical origin, cultivar, genotype

## Abstract

The seeds of 111 *Malus* sp. different fruit use (dessert and cider apples) cultivars/genotypes developed in 18 countries were analysed to evaluate composition of tocopherol homologues and identify crop-specific profile, including diploid, triploid, and tetraploid apple cultivars with and without scab-resistance to ensure high genetic diversity. The percentage of individual tocopherols was as follows: alpha-tocopherol (alpha-T) (38.36%), beta-tocopherol (beta-T) (40.74%), gamma-tocopherol (gamma-T) (10.93%), and delta-tocopherol (delta-T) (9.97%), represented by average measurements of 17.48, 18.56, 4.98, and 4.54 mg/100 g dry weight, respectively. The values of the variation coefficient showed high variability for delta (0.695) and gamma (0.662) homologue content, whereas measurements of alpha-T and beta-T were less variable (coefficient of variation 0.203 and 0.256, respectively). The unweighted pair group method with arithmetic mean (UPGMA) revealed three main cultivar groups characterised by almost equal content of all four tocopherol homologues (Group I), high concentrations of alpha-T and beta-T, but very low content of gamma-T and delta-T (Group II), and relatively high average content of alpha-T and beta-T, but higher gamma-T and delta-T content (Group III). Specific tocopherol homologues showed association with certain valuable traits, such as harvesting time (total content of tocopherols) and resistance to apple scab (alpha-T and total content of tocopherols). This study represents the first large-scale tocopherol homologue (alpha, beta, gamma, and delta) screening in apple seeds. The dominant tocopherol homologues in cultivated apple cultivars are alpha-T and beta-T, with the prevalence of alpha-T or beta-T depending on genotype. It is a unique finding due to the rare occurrence of beta-T in the plant world and is considered a unique feature of the species.

## 1. Introduction

Apples are among the most popular temperate fruit crops—the third most widely grown fruit, with global harvest reaching over 86 million tons in 2020 [1]. More than 30,000 apple cultivars are registered [2], and every year breeding programs release new cultivars to develop high-quality and disease-resistant apples. Cultivars present a broad range of apple quality attributes [3,4]. To streamline the breeding programs, it is necessary to introduce new, modern tools in apple breeding, as it is limited by the current long and complex multi-step process [4]. Phytochemicals in apples may assist the selection of *Malus* genotypes with specific nutraceutical traits suitable for establishing innovative breeding strategies [5]. One such group of compounds is tocochromanols, which have not yet been investigated or applied as potential biomarkers in apple seeds.

Tocochromanols are lipophilic antioxidants composed of a chromanol ring with varying structure and saturated or unsaturated side chains of different length. Tocochromanols include such bioactive compounds as tocopherols, tocotrienols, plastochromanol-8, and other prenyllipids [6]. Tocopherols are involved in several physiological processes of plant seeds: germination, growth, leaf senescence, responses to abiotic stresses, antioxidant function, and export of photoassimilates [7]. Although there are still many questions in plant biology related to plant resistance to biotic factors, tocopherols can be particularly important in plants as alternative defence mechanism activators under specific growing conditions [7]. Considering this, a number of studies have demonstrated the value of these lipophilic molecules in chemotaxonomy and therefore can play a significant role for taxonomic studies in some plant families [8,9,10]. For instance, tocopherol homologues can be useful biomarkers to distinguish between seeds of different oak species—*Quercus rubra* L. vs. *Quercus robur* L. [11]. Other studies confirm that tocopherols are suitable as plant functional trait biomarkers and can play an essential role in monitoring the physiological response of plants, for instance, to stress [12]. Despite significant knowledge about tocopherols, tocotrienols, and other tocochromanol functions and roles in plants, many issues associated with those unique lipophilic molecules are still unclear [7,12].

Despite many chemical composition studies on apples, few are concerned with seed tocopherols, and a limited number of genotypes has been included—only one to thirty apple genotypes were analysed [13,14,15,16,17,18]. In apple seeds, four tocopherol homologues (alpha, beta, gamma, and delta) have been found, and their proportions vary among cultivars [13]. The aim of the present study was to investigate the tocopherol homologue composition in seeds of a comprehensive number of the *Malus* sp. (dessert and cider apples) genotypes to understand the variability of these secondary metabolites and to find regularities in crop-specific profiles.

## 2. Results and Discussion

This study is the first large-scale screening of all four tocopherol homologues (alpha, beta, gamma and delta) in apple seeds. Tocopherol homologues were determined according to the previously validated RP-HPLC-FLD method [19]. All four tocopherol homologues were identified and quantified in one hundred eleven apple genotypes (Table 1). Each genotype was represented by three biological samples, ANOVA did not show statistically significant differences between biological samples or repetitions. A wide range of each tocopherol homologue concentration and high variability among tested apple samples was observed (Figure 1). The content of delta-T (0.695) and gamma-T (0.662) showed high diversity, whereas alpha-T and beta-T were less variable and more consistent among genotypes (coefficient of variation 0.203 and 0.256, respectively). The total tocopherol content ranged from 31.82 to 56.75 mg/100 g dw for cvs. ‘Mc Shay’ and ‘Kurzemes Svītrainais’, respectively. A similar range of total tocopherol concentration was reported in twelve cultivars of dessert and crab apples [13]. The mean percentage of individual tocochromanols was as follows: alpha-T (38.36%), beta-T (40.74%), gamma-T (10.93%), delta-T (9.97%), represented by mean content 17.48 (alpha-T), 18.56 (beta-T), 4.98 (gamma-T), and 4.54 (delta-T) mg/100 g, respectively.

Figure 1 presents the median value, upper and lower quartile and upper and lower extreme measurements. The bullet represents the extreme values determined for the apple cultivar ‘Uspeh’.

The dominance of alpha-T and beta-T in seeds of cultivated apples is unique in the plant world [13,14,15,16,17], and the present study. In contrast, the seeds of other pome fruits, such as pears, quince, and Japanese quince [14,20,21], or fruit crops of the *Prunus* genus [22,23,24,25] are dominated mainly by gamma-T, and quinces only contain alpha-T. The varied prevalence between alpha-T and beta-T has been observed in other studies: predominance of alpha-T over beta-T [13,15] or beta-T over alpha-T [14,16,17], which can be explained by the specificity of tested cultivars. In the current study, moderate prevalence of beta-T over alpha-T was observed in 54.1% of apple samples. The highest content of alpha-T was noted for cv. ‘Alro’ (26.29 mg/100 g dw), whereas the lowest for cv. ‘Uspeh’ (6.12 mg/100 g dw), an outlying measurement (Figure 1). The highest content of beta-T was found in cv. ‘Anīss Svītrainais’ (29.99 mg/100 g dw), the lowest—in cv. ‘Uspeh’ (6.88 mg/100 g dw) seeds. Tocopherol homologues gamma-T and delta-T had significantly lower concentrations: the highest content of gamma-T was observed in red leaf cv. ‘Carnikava’ (14.20 mg/100 g dw), whereas cv. ‘Ligol’ had the highest content of delta-T (14.05 mg/100 g dw). Cultivars ‘Katja’ and ‘Ruhm von Kirchwerder’ contained the least gamma-T (0.13 mg/100 g dw) and delta-T (0.12 mg/100 g dw), respectively. ANOVA consistently showed statistically significant differences between tested cultivars for all tocopherol homologues as well as their total content (*p* < 0.001), which allowed complete discrimination of all tested apple genotypes (Figure 2).

The analysed set of apples includes genotypes with different harvesting times: early, medium, and late (13, 67, and 31 genotypes, respectively). Using this parameter as a factor, ANOVA showed statistically significant differences between the total content of tocopherols (early apple genotypes were significantly different, average values for early—47.36 mg/100 g dw, medium—45.35 mg/100 g dw, late—45.32 mg/100 g dw), but not significant for individual tocopherol homologues. The content of tocopherols increases in seeds during fruit development in three cultivars of Japanese quince, which, same as apples, are pome fruits [26]. Based on this finding, the slightly higher total content of tocopherols in seeds of early cultivars (summer/autumn) could be explained by the higher maturity of early apple genotypes compared to medium and late varieties. Similarly, the relationship of tocopherol content and apple scab resistance was evaluated. The dataset contained 12 genotypes that were apple scab-resistant (with *Rvi5* and *Rvi6* resistance genes) and 99 susceptible genotypes. While delta-T, beta-T, and gamma-T did not show a statistically significant difference, alpha-T and total tocopherol content did. The average values for apple scab susceptible genotypes were 17.67 mg/100 g dw alpha-T and 45.85 mg/100 g dw total tocopherols content. Lower content was observed in resistant genotypes—15.75 mg/100 g dw alpha-T and 43.21 mg/100 g dw total tocopherol content. To explain the inconsistency, additional studies are needed, using material from several harvest years.

The unweighted pair group method with arithmetic mean (UPGMA), an agglomerative hierarchical clustering method based on Euclidean distances among tested apple genotypes, revealed three main cultivar groups (Figure 2), characterised by different proportions of tocopherol homologues (alpha, beta, gamma, and delta), supported by Principal Component Analysis (PCA) (Figure 3).

ANOVA of cultivar groups was performed to evaluate their reliability. Statistically significant differences were found between all analysed tocopherol homologues (*p* < 0.001) (Figure 4). Group I included six cultivars with very similar content of all four tocopherol homologues: 12.51/11.64/13.18/11.79 mg/100 g dw for alpha/beta/gamma/delta, respectively. All cultivars of this group descend from East and Central Europe (Belarus, Estonia, Latvia, Poland, and Russia) or Central Asia (Kazakhstan). In the case of Kazakhstan, the breeding parents of the analysed cultivar also had East-European origin. Like in the previous study, some cultivars (in this case, Group I) showed almost equal proportions of tocopherol homologues, for example, cv. ‘Antej’ has also been analysed previously, and an equal proportion of tocopherol homologues was observed [13]. Group II (forty-eight apple cultivars) conversely includes cultivars with very different proportions of tocopherol homologues—high concentrations of alpha-T and beta-T, very low content of gamma-T and delta-T (19.82, 22.87, 2.45, and 2.57 mg/100 g dw, respectively). The fifty-seven apple cultivars in Group III also had a comparatively high mean content of alpha-T (16.14 mg/100 g dw) and beta-T (15.69 mg/100 g dw), but higher content of gamma-T (6.45 mg/100 g dw) and delta-T (5.56 mg/100 g dw) than Group II (Figure 2). Groups II and III include cultivars of very distant origin both geographically and by parentage. Group II and III can be separated based on the proportion of alpha-T and beta-T. Although the general moderate prevalence of beta-T over alpha-T was observed and discussed above, significant differences in this parameter were observed for UPGMA-identified cultivar groups. In Group II, 22.9% of cultivars had a prevalence of alpha-T, whereas in Group III—the predominance of alpha-T was stated for 68.4% of cultivars. These observations explain contradictions in different publications about the tocopherol homologue profile in apple seed oils, discussed above. Since previous studies have analysed a relatively small number (one up to twelve) of *Malus* sp. varieties [13,14,15,16,17], they fit well into one or the other group dominated by alpha-T or beta-T. Further analysis of tocopherol content showed a close correlation between particular homologues, e.g., positive correlations: gamma-T and delta-T (r = 0.935, *p* < 0.01), alpha-T and beta-T (r = 0.676, *p* < 0.01), negative correlations: alpha-T and delta-T (r = −0.682, *p* < 0.01), alpha-T and gamma-T (r = −0.650, *p* < 0.01), gamma-T and beta-T (r = −0.654, *p* < 0.01). A similar observation was reported in acorns of *Q. robur* and *Q. rubra* [11]: positive correlations between gamma-T and delta-T as well as alpha-T and beta-T, and negative correlations between alpha-T and delta-T, alpha-T and gamma-T, beta-T and gamma-T, and, lastly, beta-T and delta-T. In the biosynthetic pathway of tocopherols, delta-T and gamma-T have two precursors: 2,3-dimethyl-6-phytyl-1,4-benzoquinol (DMPBQ) and 2-methyl-6-phytyl-1,4-benzoquinol (MPBQ), a cyclised form of MPBQ produced by methyltransferase (MT), a tocopherol cyclase. These primary products (delta-T and gamma-T) are precursors of beta-T and alpha-T, respectively, catalysed by gamma-T methyltransferase (gamma-TMT) [27,28]. Thus the obtained positive and negative correlations are related to the varying activity of enzymes involved in each of the steps of compound biosynthesis, favoring either primary or secondary products.

No relation between the calculated cultivar groups and either the harvesting time or the pedigree of the cultivars has been found.

## 3. Materials and Methods

### 3.1. Reagents

Sigma–Aldrich (Taufkirchen, Germany) provided all required solvents and reagents of analytical to HPLC grade. Standards of alpha-, beta-, gamma-, and delta-T, with purity over 95%, were obtained from Merck (Darmstadt, Germany).

### 3.2. Plant Material

The tocopherol homologues composition was evaluated in the seeds of 111 apple genotypes of selected in different geographical regions with different harvesting time, including advanced modern cultivars, selected hybrids and landraces, diverse pedigrees as well as biological and growing properties (Table 1). Genotypes included in this study were developed in 18 countries (Belarus, Canada, Czech Republic, Estonia, Finland, France, Germany, Japan, Kazakhstan, Latvia, Lithuania, New Zealand, Poland, Russia, Sweden, Ukraine, United Kingdom, and the United States), ensuring sufficient genetic diversity. A broad range of apple cultivars was analysed to evaluate the tocopherol homologue content and to find crop-specific profiles: diploid, triploid, and tetraploid cultivars, as well as cultivars with different tree structures (columnar and standard tree), leaf colour (red leaf cv. ‘Carnikava’), fruit use (dessert and cider apples), scab resistant cultivars (with *Rvi5* or *Rvi6* apple scab resistance genes). The seeds were collected in 1 September–30 November 2013 from fruits at the experimental garden/orchard (GPS location: N: 56°36′39″ E: 23°17′50″) at Institute of Horticulture, Dobele, Latvia. Three biological replicates were obtained from one to five apple trees for each genotype. If only one tree was available, apples were split into three replications. Ten to twenty apples were collected randomly from different sides of each tree. All recovered seeds from apple flesh and cores, except for those not fully developed, were used for phytochemical analysis. The seeds were immediately frozen for 24 h at −18 °C, and freeze-dried for 24 h using the Labconco FreeZone system (Kansas City, MO, USA). Lyophilised seeds were stored at −18 ± 1 °C for no longer than six months before analysis. The homogenized amount of 2 to 5 g of dry apple seeds was milled using an A 11 basic mill (IKA, Staufen, Germany). The apple seed powder (≤0.5 mm, mesh size) was immediately used for extraction of tocopherols. The solids content (dry weight; dw) of the apple seeds was measured gravimetrically. All analyses were performed at the end of 2013 and beginning of 2014.

### 3.3. Tocopherol Homologues Analysis

Samples were prepared using the same micro-saponification and extraction protocols as in previous reports [13]. Briefly, 0.1 g of powdered seeds, 0.05 g of pyrogallol, 2.5 mL of absolute ethanol, and 0.25 mL of 60% (*w*/*v*) aqueous KOH were placed in a 15 mL glass tube, and saponified (25 min, at 80 °C). Then, the sample was cooled in an ice-water bath (5 min) and 2.5 mL of 1% (*w*/*v*) NaCl solution was added and mixed (5 s). Tocopherols were extracted using 2.5 mL *n*-hexane:ethyl acetate (9:1; *v*/*v*) and mixing (15 s). The layers were separated by centrifugation (1000× *g* at 4 °C for 5 min). The organic layer was transferred to a round-bottom flask, while residues were re-extracted twice as described above. The combined extracts were evaporated in a vacuum rotary evaporator until dry, dissolved in ethanol (1 mL), filtered through a syringe filter (0.22 μm) into a vial, and injected into the RP-HPLC/FLD system (Shimadzu Corporation, Kyoto, Japan). Tocochromanols were separated in a Luna PFP column (150 × 4.6 mm, 3 μm) with a connected guard column (4 × 3 mm, 3 μm) obtained from Phenomenex (Torrance, CA, USA). The method was validated previously [19].

### 3.4. Statistical Analysis

The results were presented as means (*n* = 3). Each apple genotype is represented by three independent measurements of different biological replication of plant material. Data was processed using ANOVA. Significant differences (alpha = 0.05) were determined using the Duncan Post-hoc test for different independent samples and selected genotypes, using IBM SPSS Statistics 25 software (IBM Corp., Armonk, NY, USA, 2017). An unweighted pair group method with arithmetic mean (UPGMA) and Euclidean distance was applied to explore genotype groups using the PAST 4 (Hammer et al., 2001) software. The same software was used for principal components analysis (PCA) to identify the type of variable interaction.

## 4. Conclusions

The content and proportion of tocopherol homologues alpha-T, beta-T, gamma-T, and delta-T are highly varied, especially gamma-T and delta-T, in cultivated apple genotypes. Specific tocopherol homologues showed association with certain valuable traits, such as harvesting time (total content of tocopherols) and resistance to apple scab (alpha-T and total tocopherol content). Significant relation between the geographical origin of apple genotypes and the specific tocochromanol profile in seeds was not found. The dominant tocopherol homologues in cultivated apple cultivars are alpha-T and beta-T, with the prevalence of alpha-T or beta-T depending on genotype.

Although the alpha-T and total tocopherol content showed significant statistical differences related to scab resistance, their use for genotype evaluation can be limited by the sensitivity of the tocopherol biosynthetic pathway to various environmental factors, which can cause a false positive or false negative expectation of a certain trait.

## Figures and Tables

**Figure 1 plants-12-01169-f001:**
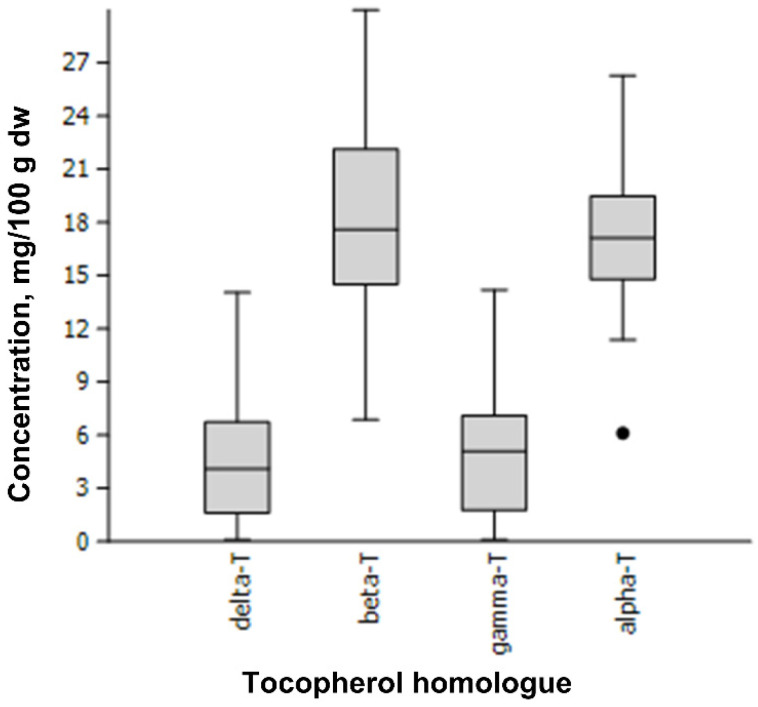
Content of tocopherol homologues (alpha, beta, gamma and delta) in 111 apple genotypes, mg/100 g dw.

**Figure 2 plants-12-01169-f002:**
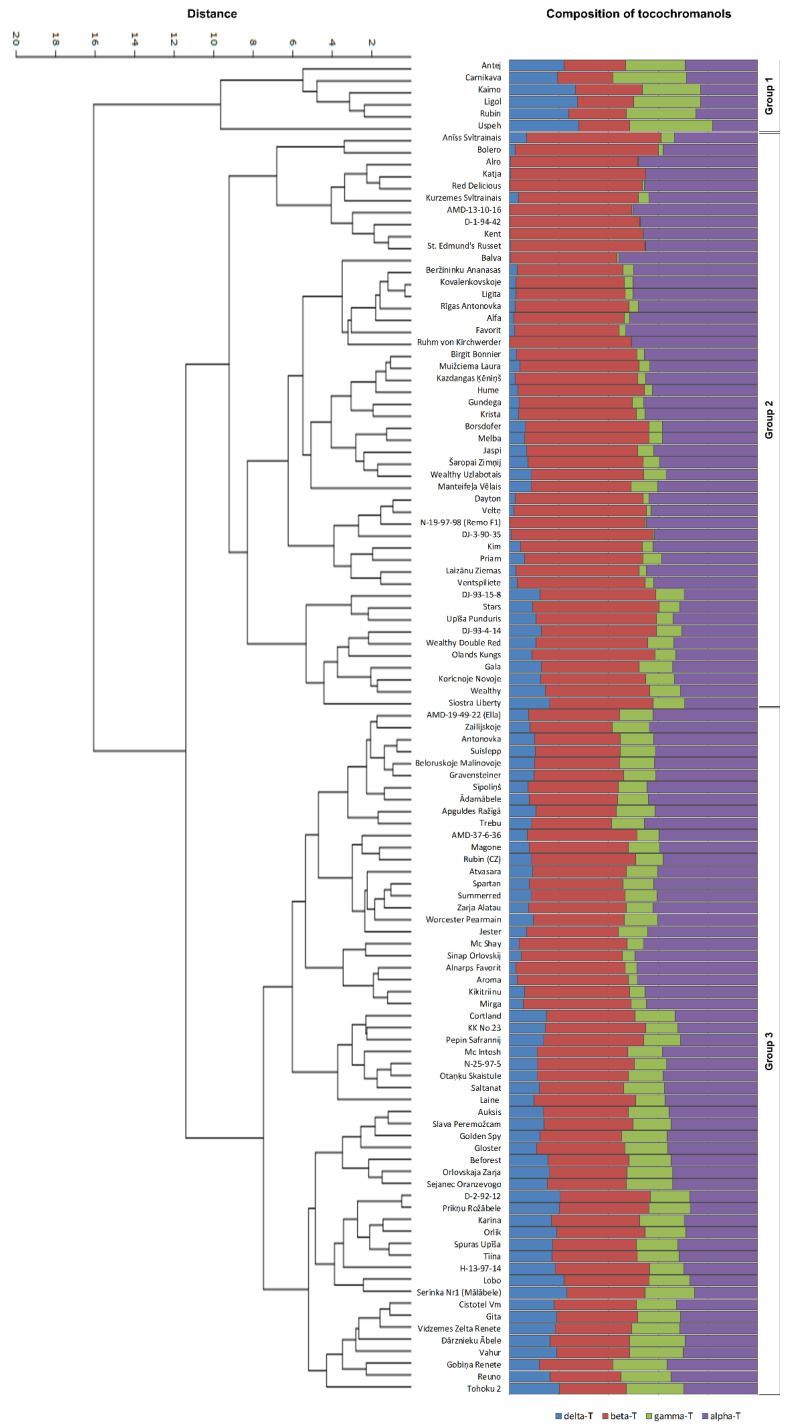
Apple genotypes grouped according to of individual tocopherol homologue (alpha, beta, gamma, and delta) content (mg/100 g dw) and their proportion (percentage, %) (Euclidean distance, UPGMA clustering).

**Figure 3 plants-12-01169-f003:**
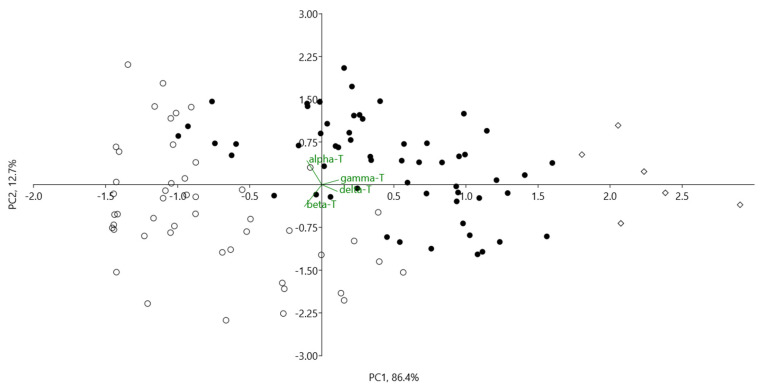
PC1 verses PC2 of 111 apples genotypes based on composition of tocopherol homologues content (alpha, beta, gamma and delta) (mg/100 g dw). Apple cultivar groups: ●—I, ○—II and ◊—III (groups identified by UPGMA cluster analysis, corresponds with Figure 2).

**Figure 4 plants-12-01169-f004:**
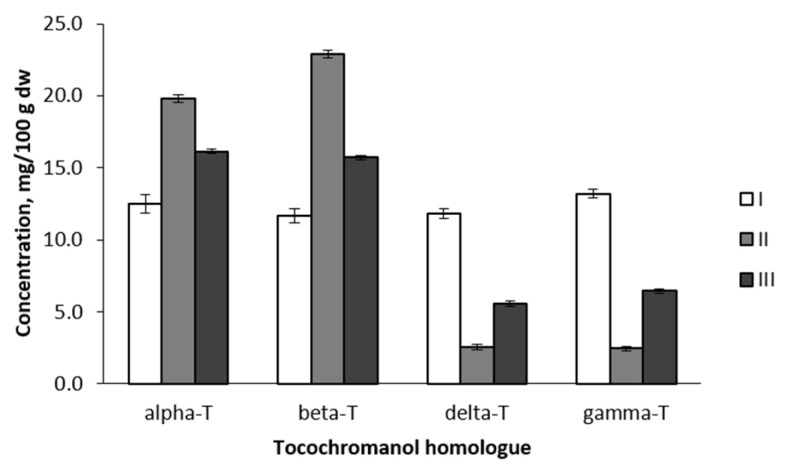
Comparison of tocopherol homologue content between apple groups identified in 111 tested genotypes. I, II and III—apple cultivar groups, identified by UPGMA cluster analysis (corresponds with Figure 2).

**Table 1 plants-12-01169-t001:** Tocopherol homologue (mg/100 g dw) contents in seeds of different various origin and harvest time cider and dessert apple genotypes.

Genotype	Country of Origin	alpha-T	beta-T	gamma-T	delta-T	Harvest Time *	Notes
Ādamābele	LVA	19.18	15.48	5.29	3.46	medium	Triploid
Alfa 68	SWE	21.81	18.66	0.93	0.82	medium	Tetraploid
Alnarps Favorit	SWE	17.64	15.96	1.70	1.01	medium	Triploid
Alro	LVA	26.29	28.00	0.15	0.30	medium	
AMD-13-10-16	LVA	24.87	24.41	0.30	0.17	late	
AMD-37-6-36	LVA	18.20	20.07	4.08	3.34	medium	
Anīss Svītrainais	RUS	18.53	29.98	2.85	3.94	early	syn. Anis Polosatij
Antej	BLR	16.45	14.12	13.57	12.42	late	
Antonovka	RUS	17.02	14.09	5.43	4.15	medium	
Apguldes Ražīgā	LVA	15.94	12.47	5.88	4.11	medium	
Aroma	SWE	18.61	17.25	1.35	1.19	late	
Atvasara	LVA	20.18	18.90	6.36	4.82	early	
Auksis	LTU	16.33	15.71	7.54	6.23	medium	
Ādamābele	LVA	19.18	15.48	5.29	3.46	medium	Triploid
Balva	LVA	22.37	17.16	0.28	0.25	medium	
Beforest	CAN	18.69	17.65	9.11	8.30	late	
Beloruskoje Malinovoje	BLR	18.26	15.07	6.06	4.52	late	
Beržininku Ananasas	LTU	23.36	19.87	2.07	1.54	early	
Birgit Bonnier	SWE	21.12	22.47	1.45	1.43	medium	
Bolero	GBR	19.52	29.50	1.01	1.32	late	Columnar, syn. Tuscan
Borsdofer	DEU	19.14	24.79	2.88	3.30	late	syn. Edelborsdorfer
Carnikava	LVA	13.54	10.56	14.20	9.17	medium	Redleaf
Chistotel	RUS	14.33	14.52	7.03	7.78	medium	*Rvi5*
Cortland	USA	12.97	13.89	6.27	5.83	late	
D-1-94-42	LVA	22.73	25.05	0.14	0.16	medium	*Rvi5*
D-2-92-12	LVA	14.17	18.72	8.18	10.65	early	*Rvi6*
Dayton	USA	18.31	21.58	0.96	0.98	medium	*Rvi6*
Dārznieku Ābele	LVA	12.50	13.67	9.55	7.11	medium	
DJ-3-90-35	LVA	15.59	21.41	0.22	0.28	late	*Rvi6*; cider apple
DJ-93-15-8	LVA	16.24	25.69	6.24	6.73	medium	cider apple
DJ-93-4-14	LVA	15.24	23.18	5.18	6.50	late	cider apple
Ella	LVA	16.25	14.22	5.11	2.98	medium	
Favorit	RUS	25.19	20.13	1.28	1.03	late	syn. Orlovskij Favorit
Gala	NZL	16.64	19.20	6.60	6.30	medium	
Gita	LVA	13.56	14.35	7.53	8.34	medium	*Rvi6*
Gloster	DEU	17.57	17.23	8.43	5.37	late	
Gobiņa Renete	LVA	15.88	12.91	9.59	5.35	medium	
Golden Spy	USA	16.66	15.17	8.51	5.64	late	
Gravensteiner	DEU	17.56	15.48	5.65	4.33	late	Triploid
Gundega	LVA	23.16	23.15	2.32	1.96	medium	
H-13-97-14	LVA	13.37	17.12	6.01	8.26	late	*Rvi6*; cider apple
Hume	CAN	20.13	24.27	1.58	1.63	medium	
Jaspi	FIN	20.69	22.14	3.02	3.45	early	
Jester	GBR	21.15	17.48	5.63	3.42	medium	
Kaimo	EST	11.38	13.20	11.56	13.19	early	
Karamba	EST	14.52	17.52	8.77	8.31	medium	syn. Karina
Katja	SWE	23.99	29.05	0.13	0.24	medium	
Kazdangas Ķēniņš	LVA	21.52	23.48	1.50	1.20	medium	
Kent	GBR	23.18	27.09	0.26	0.16	late	
Kikitriinu	EST	17.91	16.70	2.42	2.42	medium	
Kim	SWE	17.13	19.89	1.70	1.84	late	
KK No.23	EST	11.60	14.46	4.61	5.26	medium	
Korichnoje Novoje	RUS	15.95	20.35	5.58	5.95	medium	
Kovalenkovskoje	BLR	22.47	19.53	1.61	1.20	medium	
Krista	EST	23.83	24.93	1.96	1.99	medium	
Kurzemes Svītrainais	LVA	24.94	27.43	2.25	2.14	medium	
Laine	LVA	14.97	16.53	4.76	3.98	early	
Laizānu Ziemas	LVA	16.31	18.16	1.05	0.95	medium	
Ligita	LVA	22.39	19.76	1.42	1.17	late	*Rvi6*
Ligol	POL	11.81	11.75	13.93	14.05	late	
Lobo	CAN	12.75	15.87	7.66	10.26	late	
Magone	LVA	17.43	17.59	5.66	3.61	late	
Manteifeļa Vēlais	LVA	21.91	21.90	5.85	4.83	medium	
Mc Intosh	CAN	14.98	14.29	5.56	4.39	late	
Mc Shay	USA	14.67	13.92	2.02	1.23	medium	*Rvi6*
Melba	CAN	18.34	23.98	2.63	2.88	early	
Mirga	LVA	16.81	16.31	2.33	2.14	early	
Muižciema Lauka	LVA	20.71	22.86	2.03	2.09	medium	cider apple
N-19-97-98	LVA	18.48	22.51	0.20	0.16	medium	cider apple
N-25-97-5	LVA	12.77	13.60	4.52	3.94	medium	cider apple
Olands Kungs	LVA	14.68	21.85	3.75	4.14	medium	
Orlik	RUS	13.99	17.26	7.93	9.29	medium	
Orlovskaja Zarja	RUS	17.01	15.49	9.10	8.07	medium	
Otaņķu Skaistule	LVA	13.31	12.85	4.96	3.98	late	
Pepin Safrannij	RUS	12.39	16.06	5.92	5.54	late	
Priam	FRA	16.24	20.15	3.25	2.56	medium	*Rvi6*
Prikņu Rožābele	LVA	14.15	18.68	8.64	10.53	medium	syn. Roosioun
Red Delicious	USA	25.39	29.71	0.48	0.14	late	clone Cooper Zanzi Red
Reuno	EST	14.92	12.29	8.62	7.06	medium	
Rīgas Antonovka	LVA	21.52	20.58	1.69	1.07	medium	
Rubin (KAZ)	KAZ	12.16	11.25	13.66	11.78	medium	Liivi Kuldrenett × Suislepp
Rubin (CZE)	CZE	17.30	18.99	5.10	4.15	late	Lord Lambourne × Golden Delicious
Ruhm von Kirchwerder	DEU	22.83	22.10	0.16	0.12	medium	syn. Ruhm von Kirchwalden
Saltanat	KAZ	13.76	12.52	6.02	4.45	medium	
Sejanec Oranzhevogo	RUS	17.83	16.55	9.72	7.99	medium	
Serinka No.1 (Mālābele)	LVA	12.51	15.50	9.74	11.42	medium	standard clone of Serinka
Sharopai Zimņij	RUS	18.85	22.15	3.15	3.57	medium	
Sinap Orlovskij	RUS	16.90	14.13	1.76	1.73	late	syn. Orlovski Sinap
Siostra Liberty	POL	16.08	22.82	7.12	8.98	medium	*Rvi6*
Sīpoliņš	LVA	18.23	14.86	4.66	3.06	late	
Slava Peremozhcam	UKR	15.54	16.00	6.73	6.25	medium	
Spartan	CAN	19.19	17.26	5.62	3.72	late	
Spuras Upīša	LVA	16.67	17.42	8.60	9.08	early	
St. Edmund’s Russet	GBR	22.15	26.60	0.16	0.24	medium	
Stars	LVA	16.50	26.75	4.35	5.02	late	
Suislepp	EST	17.32	14.49	5.88	4.40	early	
Summered	CAN	19.21	17.94	6.14	4.29	medium	
Tiina	EST	15.75	17.17	8.41	8.54	medium	
Tohoku 2	JAP	11.72	10.74	9.08	7.95	medium	
Treboux Sämling	EST	16.50	11.66	4.83	3.26	medium	syn. Pärnu Tuviõun
Upīša Punduris	LVA	17.60	25.19	3.52	5.60	medium	
Uspeh	RUS	6.12	6.87	11.30	9.52	late	
Vahur	EST	14.37	14.04	10.24	9.10	early	
Velte	LVA	17.44	21.62	0.73	0.84	medium	
Ventspiliete	LVA	14.79	18.19	1.12	1.12	medium	
Vidzemes Zelta Renete	LVA	13.88	13.84	8.66	8.18	medium	syn. Liivi Kuldrenett
Wealthy	USA	14.57	19.91	5.84	6.82	medium	*M*. × *cerasifera* F1
Wealthy Double Red	USA	17.01	22.76	5.34	5.34	medium	Clone of Wealthy
Wealthy Uzlabotais	LVA	18.36	22.52	4.45	4.48	medium	Clone of Wealthy
Worcester Pearmain	GBR	18.32	16.64	6.17	4.49	early	
Zailijskoje	KAZ	17.37	13.26	5.95	3.29	medium	
Zarja Alatau	KAZ	19.49	18.23	4.92	3.57	late	

Country of origin is described by “Alpha-3 code” according to the ISO 3166 international standard: BLR—Belarus, CAN—Canada, CZE—Czech Republic, DEU—Germany, EST—Estonia, FIN—Finland, FRA—France, GBR—United Kingdom, JAP—Japan, KAZ—Kazakhstan, LTU—Lithuania, LVA—Latvia, NZL—New Zealand, POL—Poland, RUS—Russia, SWE—Sweden, UKR—Ukraine, USA—United States. The results are represented as the mean/average of three measurements of biological samples (*n* = 3). Unless stated otherwise in “Notes”, the genotype is a dessert apple and diploid. T, tocopherol, dw, dry weight, Vm/Rvi5, and Vf/Rvi6 are apple scab resistance genes (meaning possessing cultivars are scab resistant). * Harvest time denotes summer, autumn, and winter cultivars, respectively.

## Data Availability

The data presented in this study are available on request from the corresponding author. The data are not publicly available due to the fact that all data are based on the presented data in Table 1.

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
