# Peer review of "Tocopherols in Cultivated Apple Malus sp. Seeds: Composition, Variability and Specificity"

_plants, 2023, doi:10.3390/plants12051169_

Round 1

Reviewer 1 Report

The Methodical part is recommended to describe in more detail:  from which directions of the world (South, North, West, East) the fruits  from the apple trees were selected for the following experiments. Because it is very important the sunshine intensity for the fruit's ripeness as well as the quality of seeds.  How many apple fruits were selected from each tree?  How many seeds were prepared for the following experiments?

Author Response

Thank you for the comment and questions. The section Method of the manuscript was updated with the following information:

“The ten to twenty apples were collected randomly from different sides of each tree. All recovered seeds, except for those not developed, were used for phytochemical analysis.”

Reviewer 2 Report

This manuscript evaluates the content and composition of 111 apple genotypes coming for several countries around the world.

The document is well written and presented. Few minors corrections have to be done, and I've asked to add some discussion in a point about the earliness of the genotypes.

Please see the specific comments on the document attached

Author Response

Thank you for the comment and remarks. The manuscript has been improved according to the reviewer's guidelines.

Reviewer 3 Report

The authors did a good work from an experimental point of view, and I recommend the article for publication after some major revisions.

More specific:

L8 & L68: The sp. no italics.

L97: Why did you decide to publish the data after 10 years?

L100: Give more details about the extraction method. In a few words.

L140: Describe what you mean and which parameters you used for ripening time. Also, explain abbreviated country names.

L144: Replace the ‘’image’’ with ‘’Figure 1’’.

L185: Figure 2 in color if possible.

L189: Which method did you use for hierarchical clustering?

L193: Replace the ‘’component’’ with ‘’PC’’ in Figure 3. Also, give the percentage of each component.

Author Response

L8 & L68: The sp. no italics.

Thank you for the notification. The correction has been done.

L97: Why did you decide to publish the data after 10 years?

1st, lack of time, 2nd, for a few years the results were forgotten due to overloading and several challenges. Covid-19 and closed labs allowed us to refresh all previous results.

L100: Give more details about the extraction method. In a few words.

Thank you for the comment. The details of extraction method were provided.

L140: Describe what you mean and which parameters you used for ripening time. Also, explain abbreviated country names.

Thank you for the comment. Detailed information and explanations have been provided below the Table 1.

L144: Replace the ‘’image’’ with ‘’Figure 1’’.

Thank you for the notification. The correction has been done.

L185: Figure 2 in color if possible.

Thank you for the notification. The correction has been done.

L189: Which method did you use for hierarchical clustering?

Thank you for the question. Figure 2 states - Euclidean distance, UPGMA clustering. While in the Material and Methods part of the manuscript is provided a details: “An unweighted pair group method with arithmetic mean (UPGMA) and Euclidean distance was applied to explore genotypes' grouping using the software PAST 4...”, and Results part “The unweighted pair group method with arithmetic mean (UPGMA), an agglomerative hierarchical clustering method based on Euclidean distances among tested apple genotypes, revealed three main cultivar groups (Figure 2), characterized …”.

L193: Replace the ‘’component’’ with ‘’PC’’ in Figure 3. Also, give the percentage of each component.

Thank you for the notification. The correction has been done.

Round 2

Reviewer 3 Report

The paper has been revised according to the suggestions and criticisms of the reviewers. In this revised version, the paper has improved its quality and I recommend the article for publication.

Author Response

Thank you for your positive recommendation.